# Neuronal messenger ribonucleoprotein transport follows an aging Lévy walk

Minho S. Song[1], Hyungseok C. Moon[1], Jae-Hyung Jeon [2] & Hye Yoon Park [1,3]

Localization of messenger ribonucleoproteins (mRNPs) plays an essential role in the regulation of gene expression for long-term memory formation and neuronal development. Knowledge concerning the nature of neuronal mRNP transport is thus crucial for understanding how mRNPs are delivered to their target synapses. Here, we report experimental and theoretical evidence that the active transport dynamics of neuronal mRNPs, which is distinct from the previously reported motor-driven transport, follows an aging Lévy walk. Such nonergodic, transient superdiffusion occurs because of two competing dynamic phases: the motor-involved ballistic run and static localization of mRNPs. Our proposed Lévy walk model reproduces the experimentally extracted key dynamic characteristics of mRNPs with quantitative accuracy. Moreover, the aging status of mRNP particles in an experiment is inferred from the model. This study provides a predictive theoretical model for neuronal mRNP transport and offers insight into the active target search mechanism of mRNP particles in vivo.

[1] Department of Physics and Astronomy, Seoul National University, Seoul 08826, Korea. [2] Department of Physics, Pohang University of Science and Technology, Pohang 37673, Korea. [3] The Institute of Applied Physics, Seoul National University, Seoul 08826, Korea. Correspondence and requests for materials should be addressed to J.-H.J. (email: jeonjh@postech.ac.kr) or to H.Y.P. (email: hyeyoon.park@snu.ac.kr)

A messenger ribonucleoprotein (mRNP) particle is a macromolecular complex consisting of messenger RNA (mRNA) and RNA binding proteins (RBPs). When a gene is activated, the genetic information in DNA is transcribed into mRNA, which is subsequently associated with many RBPs to form an mRNP complex. The mRNP particles are exported from the nucleus to the cytoplasm to be translated into proteins. However, certain mRNA molecules are not immediately translated into proteins but are rather stored in a translationally repressed state and transported to specific regions inside a cell[1,2]. In neurons, about one-third of mRNA species are transported from the cell body to the dendrites and axons[3] and locally translated into proteins upon signal transduction[4–6]. Local translation of mRNA has emerged as a key mechanism for regulating gene expression in space and time for long-term memory formation and neuronal development[7,8]. Moreover, dysregulated RNA localization and translation have been strongly associated with neurodegenerative and neurodevelopmental diseases[9,10]. Therefore, it is of great interest to understand how mRNP particles are transported and targeted to specific locations inside a neuron.

Recent advances in RNA imaging techniques have enabled real-time observation of individual mRNP particles in living neurons and brain tissues[11–13]. While mRNPs in bacteria or fibroblasts exhibit mostly diffusive or subdiffusive motion[11,14,15], neuronal mRNPs show distinct directed motion along the dendrites[4,16]. Dendritic trafficking of mRNP has been shown to be mediated by microtubule-dependent molecular motors, kinesins and dyneins (Fig. 1a)[17]. A recently proposed scenario called the sushi-belt model states that mRNPs are stochastically transported along the microtubules bidirectionally until they are captured by synaptic tags at the activated synapses[18]. However, the search patterns of mRNPs as they reach their target sites are still largely unknown[4].

A Lévy walk is a physical regularization for Lévy flights that describes a generalized diffusion process in terms of a constantly moving particle of a finite velocity (in the simplest case) and the joint probability of its random flight (or jump) lengths and times[19]. The model has a finite second moment (mean squared displacement) $\langle x^2(t) \rangle$ and depending on the decay slowness of the random walker's distributions it encompasses from superdiffusion to normal diffusion with $\langle x^2(t) \rangle \propto t^\beta$ with $1 \le \beta \le 2$. Recently, interests in Lévy walks have risen within physics communities because they have been demonstrated to explain diverse complex dynamic phenomena, such as blinking quantum dots[20], cold atoms[21], light in inhomogeneous media[22], and Hamiltonian systems[23–25]. In the biological sciences, foraging movements of diverse organisms from bacteria to humans have been modeled as a Lévy process[26]. It has also been suggested that Lévy strategies are more efficient for finding randomly located target objects compared with other mechanisms[27–29]. More recently, with the help of single-particle tracking techniques[30,31], it was shown that some biological superdiffusions such as motor-driven active transport in the cytoplasm[32,33] and swarming bacteria migration[34] exhibit patterns of Lévy walks in the sense that their flight length distribution or the velocity autocorrelation decays following a (truncated) power-law.

To the best of our knowledge, this report describes the first observation of an in vivo molecular transport process with superdiffusive characteristics that are consistent with an aging Lévy walk. Theoretically, this process belongs to an extended Lévy walk model, allowing a rest period between the successive flights. Previously, such Lévy models were introduced for a flow motion in a rotating annulus and a related Hamiltonian system[24,25]. In the neuronal mRNP motion studied here, we find that this type of Lévy walk process emerges because of the bidirectional

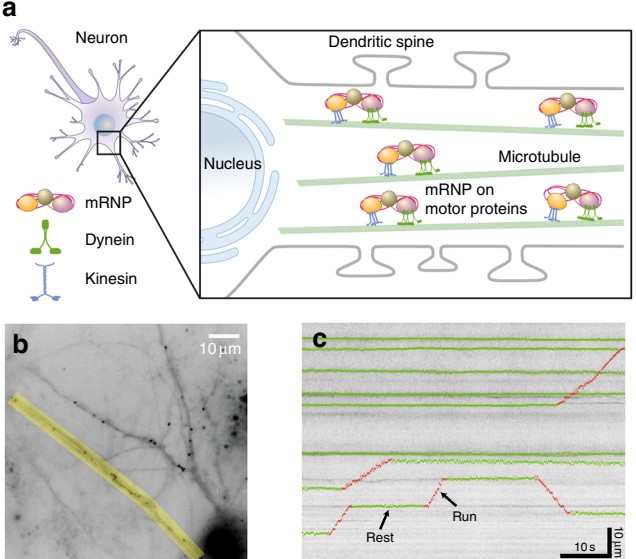

**Fig. 1** Schematic and experimental data showing motor-driven transport of mRNPs in a neuron. **a** An ensemble of mRNP complexes are transported by kinesin and dynein motor proteins along microtubules in the dendrite. In each dendrite, there are multiple microtubule tracks and thus several mRNP particles are efficiently transported at the same time to the target sites where they are localized. **b** A live-cell image showing fluorescently labeled β-actin mRNP complexes in a hippocampal neuron. Time-lapse images were taken with a time interval of $t_0 = 0.1$ s for the overall observation time of $T = 60$ s. The region of interest for analysis is marked in yellow. Scale bar, 10 μm. **c** A kymograph for an ensemble of mRNP particles obtained from an image similar to the yellow area in (**b**). Horizontal and vertical axes correspond to the elapsed time and the distance along the dendrite, respectively. For a few trajectories, their two distinct dynamic modes, rest and run, are denoted in green and red, respectively. The kymographs exhibit constant small-length scale fluctuations (see Supplementary Fig. 1 for more information). Scale bars, 10 s and 10 μm, respectively

microtubule-dependent transport by kinesin and dynein, which is interrupted by localization of mRNP at the target sites. It turns out that the dynamics of neuronal mRNP, distinguished from the pause dynamics in the tug-of-war motion of molecular motors[35,36], exhibits heavy-tailed scale-free sojourn time statistics, resulting in a sub-ballistic Lévy walk with weak ergodicity breaking (WEB) and aging. Intriguingly, aging masks WEB within the observational time window of this study. By combining single-particle tracking experiments with a theoretical study of the proposed Lévy walk process, we investigate the complicated in vivo mRNP dynamics with detailed analysis of various aspects. In the literature, Lévy-like superdiffusion has been identified mostly by the scaling analysis of the mean squared displacement (MSD) and/or the distribution profile of the flight length (or time) in a limited window. It is worthwhile to mention that beyond the simple experimental analyses, we explicitly set forth a predictive Lévy walk model for neuronal mRNP transport. Excellent agreement of this Lévy walk model with the experiment is demonstrated by various quantities, such as the ensemble- and time-averaged MSDs, the fluctuation of time-averaged MSDs, and the probability density function. Furthermore, the aging time of the observed mRNP particles in the experiment is deduced from the aging-dependent MSD curves predicted from the model. Also, the theoretical study of the Lévy model elucidates how the MSD exponents depend on the exponents of flight-localization time distributions, aging time, and noisy environment.

## Results

**Active transport of individual endogenous mRNP in neurons.** A schematic diagram of mRNP movement in a neuron is depicted in Fig. 1a. A newly transcribed mRNA binds multiple RBPs to form an mRNP, which is subsequently transported into a dendrite. There are multiple microtubule tracks along each dendrite. An mRNP particle is transported bidirectionally by concerted work of motor proteins along the microtubules. Using a genetically engineered mouse that expresses β-actin mRNA labeled with green fluorescent proteins (GFPs)[11], we imaged individual β-actin mRNP particles in hippocampal neurons using wide-field fluorescence microscopy. Because each β-actin mRNA is labeled with up to 48 GFPs, individual mRNPs appear as bright spots in fluorescence images (Fig. 1b). In neurons, mRNP complexes, often called neuronal RNA granules, contain many RBPs and ribosomes and are particularly large with an estimated size of 175–600 nm[37]. The addition of 48 GFPs, each of which is approximately 4 nm in diameter, would not significantly alter the overall size of RNA granules (less than 0.06% difference) or their native dynamics. By tracking these spots in time-lapse images, we monitored the active transport of mRNP particles in live neurons (Supplementary Movie 1). We selected a region of interest along a dendrite, as highlighted in yellow in Fig. 1b, and generated a straightened image of the dendrite. From the straightened time-lapse image stack, we obtained one-dimensional trajectories of individual mRNPs along the curvilinear line of the dendrite they reside in. Shown in Fig. 1c is one of the obtained kymographs from the selected regions of dendrites. Here, the x- and y-axes represent the elapsed time and the distance along the dendrite, respectively. We collected data from 57 cells in 12 batches of neuron culture and obtained 101 kymographs. The trajectories

of 682 mRNP particles were analyzed with a time resolution of $t_0$ = 100 ms and a spatial resolution of 205 nm.

**Transport of mRNP is composed of discrete runs and rests.** The kymographs show that the mRNP transport is intermittent and stochastic in nature. The motion of mRNP comprises two phases: namely, the directed movement, referred to as 'run' in either the anterograde or retrograde direction, and the 'rest' period (see the schematic in Fig. 2a). We segmented the kymographs into these two phases and analyzed them separately (Supplementary Fig. 1). It turns out that run is indeed a ballistic motion where the MSD grows with the power exponent of about 2 (Supplementary Fig. 2a). Meanwhile, rest is a very slow subdiffusive motion, with the exponent $\beta \approx 0.1$ for the average MSD curve (Supplementary Fig. 2b). Such seemingly subdiffusive motion for the trapped particles is ubiquitously observed in various systems, stemming from the viscoelastic anti-correlated positional fluctuation in crowded media[38–40]. In the kymographs, an mRNP particle is transported along the dendrite in such a pattern that its dynamic state is repeatedly switched between the run and rest phases with random sojourn times. Here, we note that the rest phase dominates over the run. For more than 80% of the total trajectories analyzed in the experiments, the mRNP particles were in the rest phase over the total observation time of $T = 60$ s. This long rest, presumably associated with the localization of mRNPs, appears to be a unique feature of the RNA transport in neuronal systems. This behavior is distinguished from the previously reported motor-driven active transport in cytoplasmic environments, where runs are only interrupted by short pauses with durations characterized by a sharply decaying

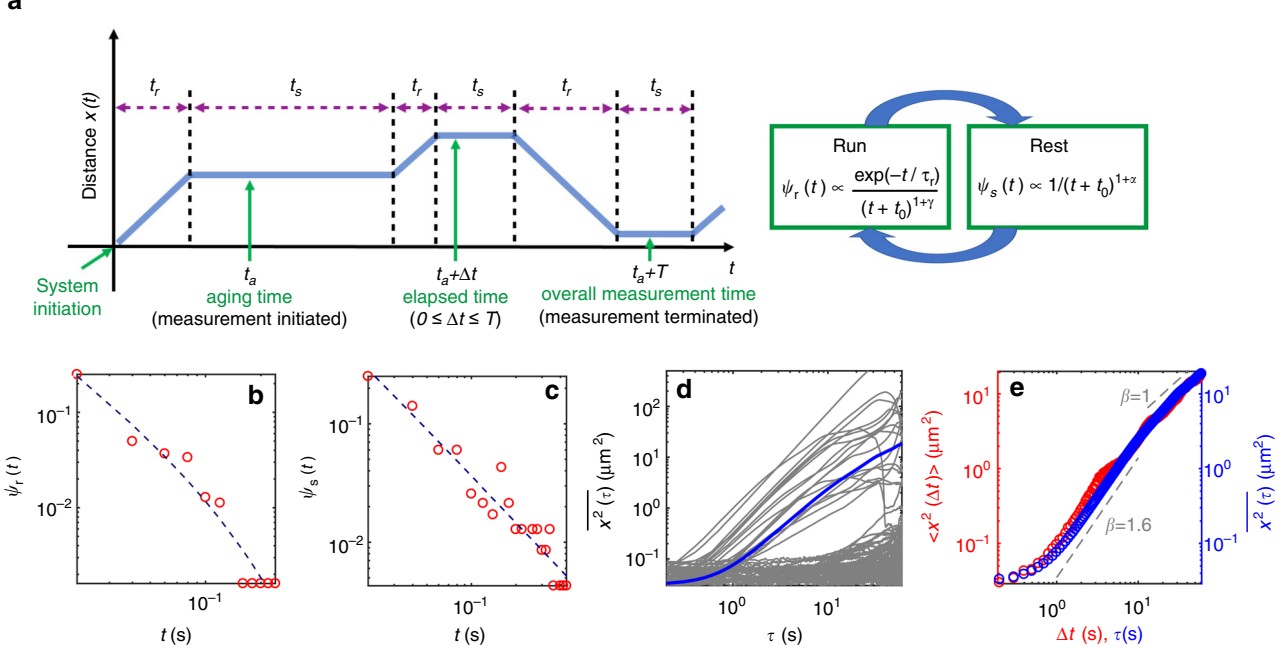

**Fig. 2** Stochastic transport dynamics of individual mRNP particles. **a** A simplified diagram describing an individual mRNP motion. Run and rest phases are repeated with randomly given sojourn times $t_r$ and $t_s$ for each. The flow diagram shows the distribution of the sojourn times. The direction of run is either anterograde or retrograde with equal probability and without memory of the previous run (Supplementary Table 1). The time $t = 0$ specifies the initial moment that an mRNP begins to be transported by motor proteins. In the experiment, the mRNP motion is observed from an arbitrary time $t_a$ (aging time) and for the overall observation time $T$. **b** Experimental data of the run time pdf $\psi_r(t)$. The dashed line represents the best fit to the data with Eq. 1. **c** Experimental data of the rest time pdf $\psi_s(t)$. The rest times follow a power-law distribution (dashed line). **d** TA MSD curves $\overline{x^2(\tau)} = \frac{1}{T-\tau}\int_0^{T-\tau}[x(\Delta t + \tau) - x(\Delta t)]^2 d\Delta t$ from individual trajectories. Thick line (blue) denotes the average curve over all TA MSDs. **e** EA MSD (red) $\langle x^2(\Delta t) \rangle = \sum_{i=1}^{N}[x_i(\Delta t) - x_i(\Delta t = 0)]^2/N$ plotted together with the average TA MSD curve (blue) shown in (**d**)

probability density function (pdf) such as an exponential decay[33,35].

We further investigated the statistical properties of both dynamic states. In Supplementary Table 1, we estimated the conditional probabilities of the four possible directional combinations for all consecutive runs separated by a rest. The four probabilities turn out to be fairly even, suggesting no directional memory in the mRNP transport. Shown in Supplementary Fig. 3a is the velocity distribution of all individual runs. The velocity profile is almost symmetric in the anterograde and retrograde directions and exhibits two sharp peaks at ~ ±1 μm/s with narrow distributions. Consistent with these data, a linear relation between the run length and time was found (Supplementary Fig. 3b). The larger error bar at longer run times is due to the smaller number of the observed events. From the curve, the average speed of mRNPs during a run period was estimated to be 1.25 μm/s. This value lies within the range of the kinesin or dynein speed experimentally estimated in vitro and in vivo[41]. In addition, we treated neurons with nocodazole to disrupt microtubules and found that the percentage of mRNPs with run phase was significantly decreased (Supplementary Fig. 4). This result suggests that intact microtubules are needed for run phase of mRNP movements.

Figure 2b, c presents experimentally determined pdfs of run and rest times, respectively. It is found that both distributions have substantially long tails; an mRNP particle is continuously transported, by a single run, for up to >20 s corresponding to distance of 20 μm, within our observation time. The run time pdf is found to be nicely explained by an exponentially truncated power-law of the form

$$\psi_r(t) \propto \frac{\exp(-t/\tau_r)}{(t + t_0)^{1+\gamma}}. \tag{1}$$

The dashed line in Fig. 2b depicts the best fit to the data with the estimated characteristic run time $\tau_r = 12.5$ s and $\gamma = 0.52$. Although the difference in the goodness of fit between the truncated power-law and a simple power-law is rather small (Supplementary Fig. 5 and Supplementary Table 2), the exponential truncation ensuring the finiteness of mRNP run times is essential in explaining the observed mRNP dynamics as described in the next section. Previously, such an exponentially decaying run distribution was theoretically elucidated within the tug-of-war model of the motor-driven cooperative transport dynamics of cargos on microtubules[35]. Further investigations on the fit functions and the goodness of the fit are discussed in Supplementary Notes 1, 2 and 3 and Supplementary Fig. 6. The pdf of run length was found to be similar to that of run time because of the relation of $l = |v|t$ (Supplementary Fig. 3c). In Fig. 2, the pdf of rest times has a much longer tail than that of run times. This is expected from Fig. 1c, in which rest events with periods exceeding $T$ are substantially found. The rest times follow a power-law decay of the form $\psi_s(t) \propto 1/(t + t_0)^{1+\alpha}$ (dashed curve in Fig. 2c), with the waiting time exponent $\alpha \approx 0.32$ (see further analyses in Supplementary Fig. 7 with Supplementary Table 3). From the fact that the rest events have a diverging average time, $\langle t \rangle = \int t\psi_s(t)\mathrm{dt} \to \infty$, it can be inferred that the overall transport dynamics of mRNP particles is highly affected by the statistics of the occurrence of long rest events.

Based on the single-trajectory analyses introduced in ref.[30], we investigated in detail the anomalous transport dynamics of mRNPs. Plotted in Fig. 2d are time-averaged (TA) MSD curves $\overline{x^2(\tau)}$ as a function of lag time $\tau$ for all individual mRNP trajectories found in the experiment (see Caption for the definition of TA MSD). Huge heterogeneity exists in individual transport dynamics, which is primarily attributable to their

localization events. The average curve over all $\overline{x^2(\tau)}$ is depicted with the thick line (blue). In Fig. 2e the more commonly used ensemble-averaged (EA) MSD $\langle x^2(\Delta t) \rangle$ is plotted as a function of elapsed time $\Delta t$ from the initiation of measurement, compared with the averaged TA MSD (blue) in Fig. 2d. Both curves, having almost the same behaviors at least up to $T = 60$ s, exhibit the following anomalous characteristics: first, superdiffusive transport of mRNPs is transient; second, they are sub-ballistic with the MSD exponent $\beta \approx 1.6 < 2$; and third, it is also notable that the short-time dynamics (of times <1 s) are not ballistic. Instead, the MSDs grow non-algebraically from a nonzero offset value at $\Delta t = 0$. As will be investigated in the simulation below, this turns out to be the consequence of positional fluctuations prevailing at short length scales, shown as a small wiggling in the kymograph trajectories (Supplementary Fig. 1). For times $>O(10)$ s, it appears that the superdiffusive motion is slowed down to a Fickian diffusion with the exponent $\beta \approx 1$. The observed trends in MSDs are not due to the inclusion of the abundant silent trajectories that correspond to a single rest event of a duration greater than or equal to the observation window $T$. Consistent scalings are found for TA and EA MSDs evaluated only with the subset of trajectories that have at least one run phase during the observation (Supplementary Fig. 8).

### Transport dynamics of mRNP follows a Lévy walk with rests.

Based on the above experimental investigation, we quantified the mRNP transport dynamics in the framework of a generalized Lévy walk model. In accordance with the experimental data, the motion of an mRNP is described as an alternating sequence of the ballistically moving run and rest phases (see Fig. 2a). In each run, the mRNP moves with a constant average velocity of $\pm v$, determined in Supplementary Fig. 3, with the equal probability 1/2. The mRNP sojourn time $t$ is given by a truncated power-law pdf (Eq. 1) with $0 < \gamma < 1$. The space-time joint pdf of the Lévy walk is then given by $\psi_{\mathrm{run}}(x, t) = \frac{1}{2}\delta(|x| - vt)\psi_r(t)$. This process is interrupted by the rest event, whose pdf, according to our observation in Fig. 2c, is modeled as a power-law function of $\psi_s(t) \propto \frac{t_0^{\alpha}}{(t + t_0)^{1+\alpha}}$ with $0 < \alpha < 1$. We emphasize that all the parameters required in this model are already determined from the experimental analyses above. In this sense the above Lévy walk is a fit-free model, enabling us to have a quantitative comparison between the experiment and the theory. Along with this truncated Lévy walk model, we considered a truncation-free Lévy walk having power-law pdfs for both $\psi_s$ and $\psi_r$ (i.e., $\tau_r \to \infty$) to quantify the effect of truncation on the transport dynamics. Note that for both Lévy walk processes, we are interested in the case where the rest times have a broader distribution than the run times, specified by the condition $0 < \alpha < \gamma < 1$ in our model. Plotted in Supplementary Fig. 9a are the sample trajectories for the generalized Lévy walk processes with and without truncation in the run time pdf. Simulations were performed with the parameter values determined from the experiment and with the initial condition $(x, t) = (0, 0)$. The two processes look evidently different even within the experimental observation time scale. For the truncated Lévy walk, the obtained trajectories (Fig. 3a) exhibit similar patterns observed in the mRNP motion (Fig. 1c); most trajectories contain rest periods longer than the overall observation time $T = 60$ s. The generalized Lévy walk including rests was previously introduced for a flow motion in a rotating annulus[24] and a related Hamiltonian system[25]. Recently, continuous time random walk models generalizing the conventional Lévy walks were studied[42]. In the context of single-particle behaviors[43,44], however, the current model has not been investigated to date.

For the proposed Lévy walk descriptions, we first investigated the properties of their MSDs. The EA and TA MSDs are plotted

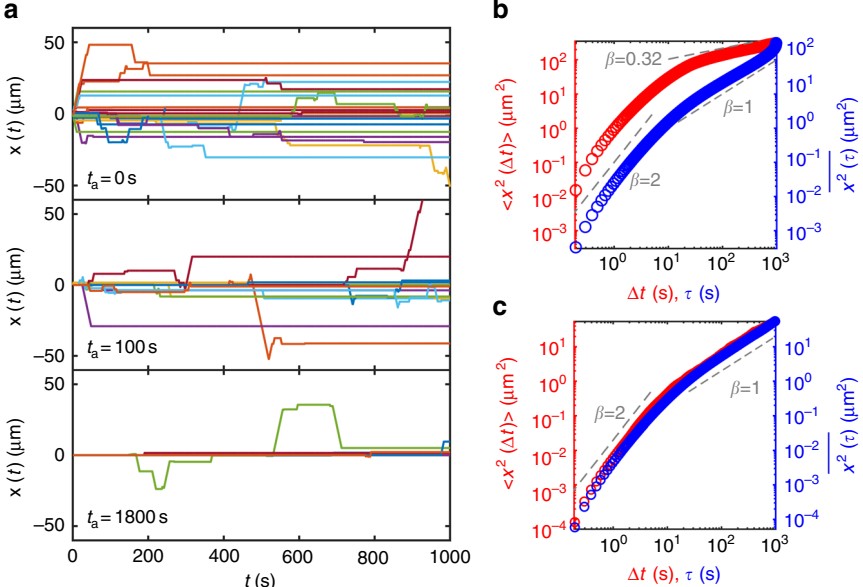

**Fig. 3** Weak ergodicity breaking and aging in the simulation of truncated Lévy walk with rests. **a** Sample trajectories from the simulations with three different measurement initiation times $t_a$ (see Fig. 2a). Top: Lévy walk trajectories when the measurement begins with the start of the process ($t_a = 0$). Middle: Trajectories when the measurement is initiated at $t_a = 100$ s. Bottom: Trajectories with the aging time $t_a = 1800$ s. In each panel, 436 trajectories are plotted, the majority of which are silent trajectories. **b** EA and TA MSD curves when the measurement initiation time is the same as the process start time (i.e., $t_a = 0$, top panel in (**a**)). Inequivalence between the EA and TA MSDs, called weak ergodicity breaking, is clearly visible in this case. For times longer than the run characteristic time, the scaling for EA MSD approaches $\sim (\Delta t)^\alpha$ while the scaling for TA MSD follows $\sim \tau^1$. **c** EA and TA MSD curves when the measurement is initiated at $t_a = 1800$ s after the start of the process (bottom panel in (**a**)). Owing to aging, the EA MSD also shows the same apparent Fickian scaling as in the TA MSD for $\tau_r < \Delta t < T$. Note that aging obscures weak ergodicity breaking in this case

together in Fig. 3b and Supplementary Fig. 9b. For the truncation-free model, both the EA and TA MSDs unanimously illustrate that the process becomes a sub-ballistic superdiffusion with the anticipated scaling exponent $\beta = 2 + \alpha - \gamma \approx 1.8$[19] soon after the short-time ballistic dynamics (Supplementary Fig. 9b). This result contradicts with the experimental curve (Fig. 2e), which shows only a transient superdiffusion with $\beta \neq 1.8$. Thus, this disagreement strongly corroborates our analysis that the run time distribution is truncated. For the truncated Lévy walk, a distinct result is observed. Importantly, its dynamics are substantially nonergodic. Both the EA and TA MSDs have the same initial ballistic growth. However, for times longer than the $\tau_r$, the EA MSD demonstrates subdiffusive dynamics of $\sim (\Delta t)^\alpha$ whereas the TA counterpart shows apparent Fickian dynamics of $\sim \tau^1$. This suggests that, depending on the method of averaging, the transport dynamics appear seemingly different. This so-called WEB for $t > \tau_r$ is mainly attributed to the intervened heavy-tailed rest events. When $t \gg \tau_r$, the truncated Lévy walk is analogous to a subdiffusive continuous time random walk governed by a waiting time pdf $\psi(t) \propto (t + t_0)^{-(1+\alpha)}$. This process is shown to exhibit WEB where the time-averaging produces $\overline{x^2(\tau)} \sim \tau / T^{1-\alpha}$ albeit the underlying motion is subdiffusive with $\langle x^2(t) \rangle \sim t^\alpha$[30,45,46]. Notice, however, that the mRNP dynamics observed in Fig. 2e are apparently inconsistent with that of the truncated Lévy walk and its WEB (Fig. 3b).

**Neuronal mRNP particles perform an aging Lévy walk.** A key point to reconcile this discrepancy is the aging effect shown for non-stationary non-equilibrium processes including the neuronal mRNP transport[30,47–49]. In such systems, the apparent dynamics explicitly depends on the aging time $t_a (>0)$ at which the observation was initiated and on the elapsed time $\Delta t$ from $t_a$ (see the schematic in Fig. 2a). We show the aging effect in our truncated Lévy walk process in Fig. 3a, where the simulated 436 truncated

Lévy walks are plotted at three different $t_a$s. With increasing aging time, the motion tends to be more immobile and fewer particles display run phases. Using the sufficiently aged trajectories ($t_a = 1800$ s), in Fig. 3c, we plot the aged EA and TA MSDs for the time window $[t_a, t_a + T]$. It shows that aging seemingly violates WEB, altering the scaling relations of the MSDs; analogously to the experimental curves (Fig. 2e), the two aged MSDs appear almost identical within the observation duration $T = 60$ s, being apparently ergodic. Moreover, the EA MSD exhibits the Fickian scaling $\langle x^2(\Delta t; t_a) \rangle \sim \Delta t$ after the ballistic regime. The observed aging behaviors may be explained by the theory of aging renewal processes[47]. In terms of the elapsed time $\Delta t$, the aged EA MSD exhibits seemingly Fickian scaling $\langle x^2(\Delta t; t_a) \rangle \sim \Delta t / t_a^{1-\alpha}$ for $\Delta t \ll t_a$ but eventually recovers the non-aged subdiffusion scaling $\langle x^2(\Delta t; t_a) \rangle \sim (\Delta t)^\alpha$ for longer observation, i.e., $\Delta t \gg t_a$. Thus, the Fickian scaling in the experimental data (Fig. 2e) is interpreted as the intermediate scaling of the aged EA MSD of the simulation data in Supplementary Fig. 10. A longer experiment would allow the observation of the terminal subdiffusive dynamics $\langle x^2(\Delta t; t_a) \rangle \sim (\Delta t)^\alpha$ as shown in the longer time scale in Supplementary Fig. 10. However, the aged TA MSD behaves as $\overline{x^2(\tau; t_a)} \sim \Lambda_\alpha \left( \frac{t_a}{T} \right) \frac{\tau}{T^{1-\alpha}}$ with $\Lambda_\alpha(z) = (1 + z)^\alpha - z^\alpha$[47], which means that the scaling relation is invariable to the aging, yet the amplitude is decreased with $t_a$ (blue curves in Fig. 3b, c). The exact same properties are observed for $\Delta t > \tau_r$ in the experiment and simulation (Figs. 2e and 3c).

At $\Delta t \lesssim \tau_r$, theoretically, the aging Lévy walk predicts the ballistic movement for mRNPs. However, a sub-ballistic motion with $\beta \approx 1.6$ is observed in the experiment (Fig. 2e). The apparent sub-ballistic dynamics are explained by considering the inherent noise from the subdiffusive motion of mRNP particles and/or the localization error in particle detection[50]. Indeed, Supplementary Fig. 1 shows that real trajectories have a small amplitude noise in both the run and rest phases. The noisy Lévy walk is constructed by superpositioning the original model with the noise extracted

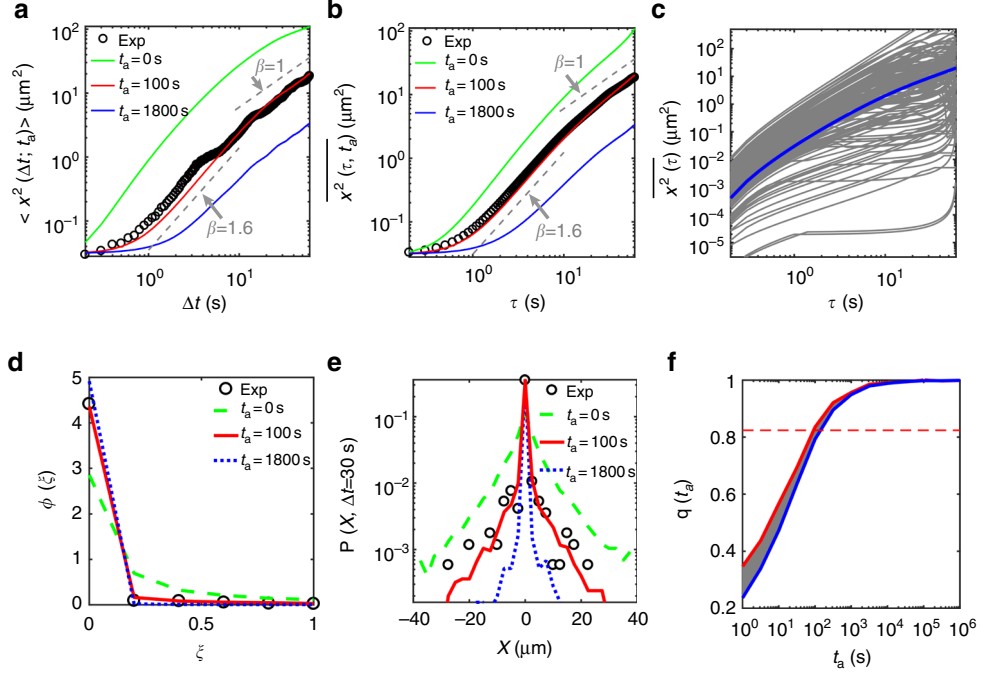

**Fig. 4** Quantitative comparison of the experiment and simulation of the aging Lévy walk process. **a** Comparison of EA MSD curves for the aging Lévy walks at various aging times $t_a = 0$, 100, and 1800 s (green, red, and blue curves, respectively) with the EA MSD from the experiment (black circles, which is the same data shown as red circles in Fig. 2e). The simulation data at $t_a = 100$ s is in excellent agreement with the experimental data. **b** Comparison of TA MSD curves. The simulation data with aging time $t_a = 100$ s, unequivocally, match both the EA and TA MSDs from the experiment. **c** Individual TA MSD curves of the aging Lévy walk processes with the aging time $t_a = 100$ s. The thick blue curve is the average of the TA MSDs (gray curves). **d** Normalized amplitude scatter distributions $\phi(\xi)$ for the TA MSDs as a function of the rescaled variable $\xi = \overline{x^2}/\langle \overline{x^2} \rangle$. The experimental distribution from Fig. 2d (black circles) is compared with the theoretically expected distributions from the aging Lévy walk at three different aging times. **e** Aged probability density functions $P(x, \Delta t)$ of mRNP particles from the experiment and the aging Lévy walk simulations. **f** A fraction $q(t_a;T)$ of the trajectories showing no run at all in $[t_a, t_a + T]$ as a function of $t_a$. Two theoretical curves for the aging Lévy walk were obtained from simulations with and without the uncertainty of ~1 μm (red and blue curves, respectively) for identifying the silent trajectories. Thus, the shaded region indicates the expected probabilities $q(t_a;T)$ in theory. The horizontal dashed line represents the experimental values for mRNP particles. It can be seen in the plot that the expected aging time is about $t_a = 100$ s

from the experimental data (see the idea of noisy continuous time random walks[51]). The effect of the noise is shown in a sample trajectory and the EA MSD in Supplementary Fig. 11. The simulation result shows that indeed the effect of the ambient noise is large enough to mask the ballistic exponent. The experimentally observed value $\beta \approx 1.6$ is seen as a crossover value from the ballistic to the Fickian exponent.

Based on the truncated Lévy walk with noise, we extracted the aging time of mRNP particles observed in our measurement. In the experiment, we obtained hundreds of mRNP traces that have different $t_a$s from the moment $t = 0$ the particle begins to be loaded and transported by motor proteins. With the assumption that the distribution for the random aging time has a well-defined mean value, we plot in Fig. 4a, b the theoretical MSDs at different aging times $\langle t_a \rangle$ for our Lévy walk process. With the average aging time $t_a \sim 100$ s, we found excellent agreement between the experiment and simulation for both EA and TA MSDs (Fig. 4a, b). Surprisingly, both their amplitudes and scalings are well explained by the simulated truncated Lévy walk process with $t_a \sim 100$ s.

We cross-checked that the aging effect indeed exists in the mRNP transport by examining several additional dynamic quantities as well as the MSD curves. Figure 4c shows the plots of the individual TA MSD curves from the simulation. The fluctuation of the TA MSDs from their average value (blue line) at a fixed lag time is displayed in Fig. 4d and Supplementary Fig. 12. Here, the amplitude scatter distributions $\phi(\xi)$ are plotted as a function of a rescaled variable of $\xi = \overline{x^2}/\langle \overline{x^2} \rangle$. The fluctuations in the experimental data are consistently explained by the

simulation data with $t_a \sim 100$ s at various lag times of 5 s (Supplementary Fig. 12a), 30 s (Fig. 4d), and 60 s (Supplementary Fig. 12b). The sharp peak at $\xi \sim 0$ signifies a portion of trajectories having no displacement, the population of which is increased with $t_a$. This effect has been coined as population splitting, and was first recognized in ref.[47] (see also Supplementary Fig. 13). In Fig. 4e, we compare the aged probability density functions $P(x, \Delta t)$ of mRNP particles determined from the experimental and simulation data. We also found excellent agreement between the experiment and simulation with $t_a \sim 100$ s, with both having a sharp cusp at the origin. Finally, we estimated the probability $q(t_a; T)$ of finding a silent trajectory (i.e., no run event at all) in $[t_a, t_a + T]$ as a function of $t_a$ (Fig. 4f)[47]. As expected from Fig. 3a, this probability increases with $t_a$, eventually saturating to unity. Figure 4f depicts the predicted $q(t_a;T)$ from the simulation in the presence of the experimental uncertainty of ~1 μm for discriminating rest (i.e., the silent trajectory) from run. From the experiment, $q$ was estimated to be approximately 0.82 (the horizontal dashed line) for the overall observation time $T = 60$ s. Again, the expected aging time for $q \approx 0.82$ turns out to be $t_a \sim 100$ s. Further analyses on the agingness in the mRNP transport are provided in Supplementary Figs. 14 and 15.

What, then, is the meaning of the measured aging time? Recall that $t_a$ is effectively the elapsed time starting from when mRNP particles had their first run to when the measurement is performed. Accordingly, the estimated aging time provides information about the effective distance $\sim \langle x^2(t_a) \rangle^{1/2}$ of mRNP particles from the cell body (dendrite entrance) when they were measured in the experiment. According to the theoretical EA

MSD curve (Fig. 3b) of our Lévy walk, the traveled distance for the elapsed time $\Delta t = 100$ s is approximated to be ~10 μm. The estimated distance is indeed consistent with our experiment where the mRNP data for kymographs were taken in the proximal dendrite of length 0–50 μm from the cell body (Fig. 1b). The aging property presented here is a purely physical effect arising from the prevailing long rest events responsible for mRNA localization dynamics. Although the full microscopic mechanism underlying mRNA localization is still unknown, the heavy-tailed rest time distribution may be associated with the dynamic nature of synaptic tags and their interactions with mRNP particles. For instance, the dynamic interaction between clathrin-coated pits and endocytic cargos also displays long-tail power-law statistics[52]. A similar effect may be present between synaptic tags and target mRNPs, but the molecular composition of synaptic tags still needs to be identified. Additionally, regarding the microscopic mechanism for the exponentially truncated run distribution, our control experiment demonstrated that run is a microtubule-dependent directed motion (Supplementary Fig. 4). It is speculated that the exponential truncation is related to the microtubule-associated kinesin/dynein-driven transport in a tug-of-war manner[35].

## Discussion

In this study, we provide a systematic investigation on the stochastic transport and localization dynamics of neuronal mRNP particles in vivo. Their combined dynamics of individual mRNP particles follows a generalized aging Lévy walk comprised of a ballistic run with a truncated Lévy distribution of the sojourn times and a localizing rest with a Lévy distribution. The aging dynamics, distinguished from the usual Lévy-like superdiffusion in other biological systems[32–34], turns out to be a prominent feature of the neuronal mRNP transport having a broad spectrum of localization time scale. Upon key dynamic observables such as the scaling of EA and TA MSDs, fluctuations in TA MSD, aged probability density, and fraction of silent trajectories, our aging Lévy walk model demonstrates excellent agreement with the experiment as well as knowledge of the proper interpretation of nontrivial aging mRNP dynamics. While there have been a few theoretical models to predict the optimal parameters for mRNP trafficking in neurons[36,53,54], our study is the first to suggest a predictive theoretical model and to perform an extensive quantitative comparison between the model and experimental data.

We envision that the aging Lévy walk model can serve as a framework for characterizing the transport dynamics of different cargos in neurons. It remains unknown how different mRNA molecules are packaged into distinct mRNP granules to be delivered to their own target sites. Likewise, other cargos, such as mitochondria, neurotransmitter receptors, and endoplasmic reticulum subcompartments, could exhibit different transport characteristics. With more detailed information on transport and localization dynamics, a next step would be to evaluate the efficacy of the aging Lévy walk as an optimal search mechanism for hidden target sites.

## Methods

**Neuron culture**. All animal experiments were performed in accordance with the protocols approved by the Institutional Animal Care and Use Committee at the Seoul National University. Mouse hippocampal neurons were isolated from 0- to 2-day-old transgenic mouse pups that express GFP-labeled β-actin mRNA as described previously[11]. Dissociated hippocampal neurons were plated onto poly-D-lysine (Sigma-Aldrich, P7886) coated glass-bottom dishes (MatTek). Cultures were maintained in Neurobasal-A medium (Gibco, 10082–147) supplemented with B-27 (Gibco, 17504–044), Glutamax (Gibco, 35050–061), and Primocin (Invitrogen, ant-pm-1) at 5% $CO_2$ and 37 °C for 14–16 days before imaging.

**Live-cell imaging**. For live-cell imaging, culture medium was replaced with pre-warmed (37 °C) HEPES-buffered saline (HBS) (119 mM NaCl, 5 mM KCl, 2 mM

$CaCl_2$, 2 mM $MgCl_2$, 30 mM glucose, 20 mM HEPES at pH 7.4). Wide-field fluorescence images were acquired using an Olympus IX-71 inverted microscope with UApo 150×1.45 NA oil immersion objective lens, an iXon electron-multiplying charge-coupled device, and an MS-2000 XYZ automated stage. During imaging, neurons were maintained at 37 °C and 60% humidity in an incubation chamber on the microscope. A 488 nm argon ion laser was used for the excitation of GFP. A 525/30 bandpass filter (Semrock) was used to detect the emission from GFP. Time-lapse images were taken in a single plane at 10 fps using streaming acquisition mode in MetaMorph software. For nocodazole experiments, the same dendrites were imaged first in HBS and after the treatment with 5 μg/ml nocodazole for 4 h.

**Image analysis**. To analyze mRNP movements in a neuron, we selected a region of interest along a dendrite, straightened the dendritic segment, and generated a kymograph using Fiji software as described previously[55]. For sub-pixel localization of particles in kymographs, we detected local maxima and calculated the one-dimensional centroid of each particle using custom MATLAB code. Each trajectory in a kymograph was automatically segmented into run and rest phases according to the local velocity along the trajectory. At each time point, the local velocity $v(t)$ was determined using a linear fitting of 11 data points from $x(t-5)$ to $x(t+5)$. If all three local velocities, $v(t-5)$, $v(t)$, and $v(t+5)$, were smaller than 0.5 (equivalent to 0.5335 μm/s), $x(t)$ was regarded to be in the rest phase. If any one of the three local velocities were larger or equal to 0.5, $x(t)$ was regarded to be in the run phase. The automatically segmented run and rest phases are indicated in red and green, respectively, as shown in Fig. 1c. This algorithm was able to successfully distinguish between run and rest phases in more than 95% of the trajectories that we analyzed. If the automatic segmentation did not work, we manually assigned run and rest phases in less than 5% of the trajectories. The global velocity of each run phase shown in Supplementary Fig. 3a was determined by dividing the total travel distance by the run time.

**Numerical simulations**. We simulated our generalized Lévy walk processes, schematically depicted in Fig. 2a, in continuous space. Two types of models were explicitly considered in the study. The truncation-free model: Both pdfs of run and rest times are of power-laws and their normalized pdfs are, respectively, $\psi_{run}(t) = \frac{\gamma}{(t/t_0+1)^{1+\gamma}}$ and $\psi_{rest}(t) = \frac{\alpha}{(t/t_0+1)^{1+\alpha}}$. The truncated model: In this case, while the $\psi_{rest}$ remains the same power-law, the run time pdf is an exponentially truncated power-law, $\psi_{run}(t) = \frac{t_r^\gamma}{e^{1/t_r}\Gamma(-\gamma,t_r^{-1})}\frac{e^{-t/t_r}}{(t/t_0+1)^{1+\gamma}}$. In the simulations, the experimentally estimated parameter values, $\gamma = 0.52$, $\alpha = 0.32$, and $t_r = 12.5$ s were consistently used for both models and the unit time was set to $t_0 = 1$ s. The random numbers were generated by the Mersenne Twister method[56] (based on the C language). The unit length in the simulation was set to be the speed of a Lévy walk during the run phase equal to the experimentally obtained average value $v = 1.25$ μm/s. For all simulations, the process started at $t = 0$ s with run and the trajectories were recorded in the time window $[t_a, t_a + T]$ for the arbitrary aging time $t_a$ with the sampling time interval 100 ms and the observation time $T = 60$ s. To obtain MSD curves, at least 10,000 trajectories were produced in each case.

**Data availability**. The data sets generated and analyzed during the current study are available in the figshare repository, doi:10.6084/m9.figshare.5549290.

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

## Acknowledgements

This research was supported by Samsung Science and Technology Foundation under Project Number SSTF-BA1602-11 to H.Y.P. and Basic Science Research Program through the National Research Foundation of Korea (NRF) funded by the Ministry of Science, ICT & Future Planning (No. 2017R1C1B2007555 & 2017K1A1A2013241) to J.-H.J. We thank Ralf Metzler and Hyun-Woo Lee for critical feedback on the manuscript. We also thank Vasily Zaburdaev for helpful discussions.

## Author contributions

J.-H.J. and H.Y.P. designed the study. H.Y.P. performed experiments. J.-H.J. developed theoretical models. M.S.S. analyzed the experimental data. H.C.M. performed simulations. All authors have discussed the results and wrote the paper.

## Additional information

**Competing interests:** The authors declare no competing financial interests.

