## [Peer Review File · Nature Communications]

Reviewers' comments:

Reviewer #2 (Remarks to the Author):

In this paper, the authors study the transport dynamics of neuronal mRNPs, which plays an essential role in neural system functions. Based on experimental measurements of single-mRNP transport in living neurons and theoretical analysis of transport trajectories, the authors propose that the transport follows a generalized Levy walk with two alternating phases, i.e., a ballistic run phase, whose time interval follows a truncated power-law distribution, and a rest phase, whose time interval follows a standard (non-truncated) power-law distribution. The authors show that the model can quantitatively reproduce the key dynamic characteristics of mRNP transport, including the mean squared displacement (MSD) of the transport trajectory. By comparing the time-averaged (TA) and ensemble-averaged (EA) MSDs, the authors further propose that the transport dynamics is nonergodic with aging effect, which agrees with experimental results.

I think this is a potentially very important finding, which provides us a predictive power for the neuronal mRNP transport process, and may help us understand the target search mechanism of mRNPs in vivo. The manuscript is well written. I only have a few minor concerns:

(1) In Fig. 2, to help the readers understand the key points of the authors' model, it may be better to have an additional panel between Panels c and d. For example, the authors may plot a circular flow diagram to show the transitions between the run and rest phases, and mark on each phase the distribution of the corresponding time interval.

(2) To find out the expression of the run-time distribution, the authors try four types of functions. However, why those four types are chosen is not discussed. Also, to determine which type of function gives the best fit to the experimental data, the authors compare the R^2 and RMSE of the fits. However, since the values of R^2 and RMSE from different fits are very close, such comparison may not be convincing enough. For instance, the experimentally measured run-time distribution (red circles in Fig. S10) seems to show a plateau at $\Psi_r \sim 10^{-1}$, which, if real, will favor the double-exponential fit. On the other hand, if we trust the truncated power-law fit, the plateau should be noise. Without further experimental data, it is hard to judge which hypothesis is true. Hence, the authors may want to collect more experimental data to lower the noise level. If there is difficulty running more experiments, the authors may discuss other evidence to favor the truncated power-law function. For example, as the authors mention in Page 3, the function has been "elucidated within the tug-of-war model of the cooperative motion for motor proteins". Is this type of motion related to mRNP transport? Moreover, to be more comprehensive, the authors may also discuss how the other three function forms, if used, will affect the transport behavior (e.g., the shape of MSD curve).

(3) At the end of Page 4, to reconcile the discrepancy between Figs. 2e and 3b, the authors introduce the aging effect, which is then demonstrated and compared with experimental data using numerical simulation. To be more comprehensive and more convincing, the authors may consider the following experimental-data-based test: From all trajectories, create a subset containing those that have at least one run phase during the observation. For each trajectory in the subset, reset the time axis so that the observation starts from the first time frame of the first run phase. After this "alignment" step, re-calculate EA MSD and compare with TA MSD. If the aging effect exists, the above operation should reverse it. Hence, we are supposed to see from the realigned experimental data the inequivalence between the EA and TA MSDs as in Fig. 3b.

(4) In the Discussion part of the paper (at the end of Page 7), besides discussing the microscopic mechanism of the rest-time distribution, the authors may also want to discuss the microscopic

mechanism of the run-time distribution.

(5) According to the Instructions of Nature Communications (https://images.nature.com/full/nature-assets/ncomms/pdf/submission_guide_ncomms.pdf), the authors should provide a Data Availability statement in the Methods section under "Data Availability".

Reviewer #3 (Remarks to the Author):

Report of Referee

In this paper the Authors report experimental and theoretical evidence that the active transport dynamics of hippocampal neuronal mRNP's follows an aging Levy walk.

Suppose we begin observations of a system which was initialized at $t = 0$ after some time t_a . There can be many reasons why we would want to do this, from technical restrictions of a measuring device to sheer curiosity. In many cases the delay largely changes statistical properties of the observed process. Such a phenomenon is called aging, a term which was originally used in the area of glassy materials [1] was also reported for blinking nanocrystals [2].

In such systems statistical properties (like probability distribution, mean square displacement, first-passage time) depend on a time span t_a between the initialization and the beginning of observations.

For example, in a paper [3] aging properties of ballistic Levy walks and two closely related jump models: wait-first and jump-first were studied theoretically. Moreover, their probability distributions and MSDs were explicitly calculated. It turns out that despite similarities these models react very differently to the delay t_a . Aging weakly affects the shape of probability density function and MSD of standard Levy walks.

Here, Authors clearly demonstrated that their model of Levy walk including rests reproduces the experimentally extracted key dynamic characteristics of mRNPs with quantitative accuracy. Moreover, this predictive theoretical model offers insight into the active target search mechanism of mRNP particles in vivo.

The paper, as well as, Supplementary Information are very well written and statistically correct. According to my knowledge the paper should be interesting to a wide readership since the main conclusion is the following: the proposed here new model in the form of aging Levy walk

is well fitted to the extensive experimental data and distinct from the previously reported in the literature motor-driven transport.

Below are a two specific comments for the consideration of the Authors.

1/ The Authors should compare their model with those studied in paper [3]. For example, the empirical density presented on Fig.4 e corresponds to the general theoretical result presented on Fig. 4 in [3] for aging jump first Levy walk ? In the manuscript Authors compare only experimental data with simulations of the fitted truncated Levy walk with rests, see Fig.3a. Following [3] one can compare the data with known theoretical density of the proper aging Levy walk.

2/ The Authors studied weak ergodicity breaking of their data see Fig.3.b and Fig.3C by comparing TA MSD with EA MSD. Their conclusions are right. Similar problem was studied recently in [4], where the authors present the experimental and data analysis investigation of the weak ergodicity breaking phenomenon revealed in the dynamics of membrane proteins on the somatic surface of hippocampal neurons. A different statistical analysis and tools for the determination of ergodic properties of single particle trajectories were used there.

[1] E. M. Bertin and J. P. Bouchaud, Phys. Rev. E 67, 026128 (2003).

[2] X. Brokmann, J.-P. et al. Phys. Rev. Lett. 90, 120601 (2003).

[3] M. Magdziarz and T. Zorawik, Phys. Rev. E 95 022126 (2017).

[4] D. Krapf et al., Scientific Reports 7, 5404 (2017).

Under the above facts, I strongly support the publication of this material in Nature Communications, after the Authors reply to the above remarks and possible improvement points above (minor revision). The latter are, however, mainly related to the presentation of the results.

Reviewer #4 (Remarks to the Author):

The study by Song et al. examines the motion of b-actin mRNPs in living neurons and sets out to assign a physical model to their type of motion. The proposed model is called Levy walk. It would have been helpful if they would have tested some inhibitors of relevant motors or disassemble microtubules, to give biological relevance to the study.

Citations of key studies are missing, such as: Yoon et al. Glutamate-induced RNA localization and translation in neurons. PNAS 2016.

Journal: Nature communications

Manuscript: NCOMMS-17-22139-T

Title: Neuronal mRNP transport follows an aging Lévy walk

Authors: Minh S. Song, Hyeongseok C. Moon, Jae-Hyung Jeon, and Hye Yoon Park

REPLY TO THE REFEREES

We appreciate the Reviewers' constructive comments on our manuscript. Below, we reply to their questions and comments point by point. All the changes/additions made in the manuscript & Supplementary Information are marked with red color.

Reviewer #2

In this paper, the authors study the transport dynamics of neuronal mRNPs, which plays an essential role in neural system functions. Based on experimental measurements of single-mRNP transport in living neurons and theoretical analysis of transport trajectories, the authors propose that the transport follows a generalized Levy walk with two alternating phases, i.e., a ballistic run phase, whose time interval follows a truncated power-law distribution, and a rest phase, whose time interval follows a standard (non-truncated) power-law distribution. The authors show that the model can quantitatively reproduce the key dynamic characteristics of mRNP transport, including the mean squared displacement (MSD) of the transport trajectory. By comparing the time-averaged (TA) and ensemble-averaged (EA) MSDs, the authors further propose that the transport dynamics is nonergodic with aging effect, which agrees with experimental results. I think this is a potentially very important finding, which provides us a predictive power for the neuronal mRNP transport process, and may help us understand the target search mechanism of mRNPs in vivo. The manuscript is well written. I only have a few minor concerns.

Authors' reply:

We would like to thank the Reviewer for the overall positive evaluation of our manuscript and for the constructive comments. Our responses to the comments are presented below.

Comment 2-1:

In Fig. 2, to help the readers understand the key points of the authors' model, it may be better to have an additional panel between Panels c and d. For example, the authors may plot a circular flow diagram to show the transitions between the run and rest phases, and mark on each phase the distribution of the corresponding time interval.

Authors' reply:

Following the Reviewer's suggestion, we added a circular flow diagram in Fig. 2a in the revised manuscript, and also described it in the figure caption.

[Fig. 2 caption in Page 4, Main text]

■ Added sentence

The flow diagram shows the distribution of the sojourn times.

Comment 2-2:

To find out the expression of the run-time distribution, the authors try four types of functions. However, why those four types are chosen is not discussed. Also, to determine which type of function gives the best fit to the experimental data, the authors compare the R^2 and RMSE of the fits. However, since the values of R^2 and RMSE from different fits are very close, such comparison may not be convincing enough. For instance, the experimentally measured run-time distribution (red circles in Fig. S10) seems to show a plateau at $\Psi_r \sim 10^{-1}$, which, if real, will favor the double-exponential fit. On the other hand, if we trust the truncated power-law fit, the plateau should be noise. Without further experimental data, it is hard to judge which hypothesis is true. Hence, the authors may want to collect more experimental data to lower the noise level. If there is difficulty running more experiments, the authors may discuss other evidence to favor the truncated power-law function. For example, as the authors mention in Page 3, the function has been “elucidated within the tug-of-war model of the cooperative motion for motor proteins”. Is this type of motion related to mRNP transport? Moreover, to be more comprehensive, the authors may also discuss how the other three function forms, if used, will affect the transport behavior (e.g., the shape of MSD curve)

Authors’ reply:

We thank the Reviewer for pointing out this subtle issue that we missed to discuss sufficiently in the manuscript. In the revised version, we tried to clarify this issue by following the suggestions made by the Reviewer. In the Supplementary Information (SI page 2-3), the section of fitting run distribution is significantly expanded to discuss the physical grounds for the choice of the four fit functions (SI page 2), and to investigate the effects of fit functions on the transport dynamics (SI page 3) with an additional supplementary figure showing new theoretical MSD curves for the four fit functions (Supplementary Fig. 15).

[Line 27 to 36 in Page 2, Supplementary text]

■ Added sentences

The run time distribution shown in Fig. 2b was fit with the following distribution functions. The three variations of the exponentially decaying function (single-exponential, truncated power-law, and double-exponential) were chosen based on a simple physical argument that the run has a finite characteristic length in that there should be a finite number of ATPs involved in a single run event. This idea is also corroborated by a theoretical study by Müller et al.¹, which showed that the transport dynamics of a cargo via a tug-of-war competition by multiple motors leads to an exponentially truncated distribution of run length/time at large length/long time. On

the other hand, run and rest occurring times can be viewed as bursting times in a complex system, where commonly observed bursting time probability density function (pdf) is of a power-law². Accordingly, below we consider the four distinct distribution functions.

[Line 54 to 76 in Page 3, Supplementary text]

■ Added sentences

Effects of the profile of distribution functions on the transport dynamics

As pointed out in the main text and Supplementary Fig. 8, the transport dynamics of a generalized Lévy walk process (modeling the mRNP motion) is evidently distinct depending on whether the exponential truncation exists or not in the power-law pdf of the run time. However, it turns out that for the observation time window in our experiment, the difference among the above three variations of the exponentially decaying functions is negligible in the transport dynamics as shown in Supplementary Fig. 15. The ensemble- and time-averaged MSDs of our generalized Lévy walk model for the above four fit functions are plotted with their best fit parameters in Supplementary Fig. 15. The results demonstrate that the three fit functions in the class of exponential function display very similar patterns of the ensemble- and time-averaged MSDs. This further augments our conclusion above (and in the main text) such that the exponential truncation is critical in characterizing the mRNP dynamics in our observation time scale while the details of the exponential pdf is irrelevant if it fits sufficiently well the data within the observation time window. The observed robust scaling behavior is physically understandable; at short times the dynamics is dominated by the run phase leading to the ballistic scaling while the exponent of long-time dynamics ($>$ the truncation characteristic time) is determined by the waiting time pdf of rest events. The curvature around the cross-over time and the amplitude of the MSD curves may, in general, depend on the details of the (exponential) distribution function. Even in this case, the exponential pdfs tuned with a proper normalization and fit parameters within given time window can produce similar cross-over characteristics. Finally, we note in passing that the amplitude of the MSD curve can vary depending on whether the part of pdf whose time is shorter than our fitting range (> 2 s) is considered or not for normalization (data not shown). In this study, we used exponentially truncated power-law distribution, which is conceptually easy to handle in the framework of Lévy walk theory.

In the main text (page 3), we also added a few sentences explaining a new control experiment resolving the issue whether the active motion of mRNP microscopically originates from motor-driven motion along microtubules (see Supplementary Fig. 4). Also, on page 3, we revised the text explaining the run time distribution to emphasize the importance of the exponential truncation in the probability density function (pdf).

[28th to 33rd line, 2nd column in Page 3, Main text]

■ Added sentences

In addition, we treated neurons with nocodazole to disrupt microtubules and found that the percentage of mRNPs with run phase was significantly decreased (Supplementary

Fig. 4). This result suggests that intact microtubules are needed for run phase of mRNP movements.

[45th to 54th line, 2nd column in Page 3, Main text]

■ Added sentences

Although the difference in the goodness-of-fit between the truncated power-law and a simple power-law is rather small (Supplementary Fig. 5 and Supplementary Table 2), the exponential truncation ensuring the finiteness of mRNP run times is essential in explaining the observed mRNP dynamics as described in the next section. Previously, such an exponentially decaying run distribution was theoretically elucidated within the tug-of-war model of the motor-driven cooperative transport dynamics of cargos on microtubules [38].

Additionally, in this Reply, we provide a few supplementary remarks regarding this comment: (1) Originally, when we began to analyze the run statistics in view of the Lévy walk theory, the primary concern was to judge whether the run time pdf follows a simple power-law or not. This issue was raised because of the fact that in literature, often, the limited experimental data is treated with a power-law statistics and associates the corresponding dynamics with a Lévy walk. Although the truncated power-law was a better fit function to the data in our study, as the Reviewer pointed out, R^2 and RMSE did not show a large difference between the truncated power-law and a simple power-law. In the current work, we resolved this issue with the help of a Lévy walk theory. Even though within the experimental limitation it is hard to differentiate a truncated power-law from a truncation-free one, it turned out that the theoretical mRNP dynamics obeying the power-law run statistics should produce a permanent superdiffusive motion. This contradicts with our experiment and theory (for the truncated pdf) which show the Fickian scaling at long times (Supplementary Fig. 8 & the comments in the main text (1st to 18th lines, 2nd column in page 5 and 6th to 15th lines, 1st column in page 6)). Although the difference in the fit goodness was small, we are convinced that the existence of exponential truncation in run time pdf was sufficiently explained in the manuscript.

(2) While there exists huge difference in the transport dynamics between the cases obeying the truncation and truncation-free run pdfs, the difference among the variations of the exponentially decaying function (namely, the exponential, exponential power-law, and the double exponential) is negligible if they are tuned with a proper normalization and fit parameters to explain the data within an observation time window. See further explanation on this issue with a new simulation result in Supplementary Information (6th to 27th line in page 3,

Supplementary Text and Supplementary Fig. 15).

(3) Based on the fit goodness, the exponentially truncated power-law was chosen in this study. We also would like to remark that the exponentially truncated power-law, *a.k.a* the tempered power-law distribution is conceptually easy to handle in the framework of Lévy walk theory. This function interpolates the conventional (genuine) Lévy walk dynamics at times shorter than the exponential characteristic time and the truncated behavior at times longer than this. In the former case, its dynamic properties were extensively studied with analytic results in literature (see, e.g., Zaburdaev et al. Ref. [19]).

(4) A widely accepted picture of the neuronal particle transport is that mRNP particles are moved by kinesin and dynein motor proteins on microtubules. In the revised manuscript, we presented a new experiment favoring this scenario (28th to 33rd line, 2nd column in page 3 of the main text and Supplementary Fig. 4). The general theory on the tug-of-war dynamics of cargo transport by multiple motors on microtubules, studied by Müller et al. (Ref. [38]), predicts essentially an exponentially decaying run distribution at large length/time. The exponential truncation in the run time pdf in our study is consistent with this picture. To explain this more clearly, the corresponding sentence in the main text was slightly modified as below.

[51st to 54th line, 2nd column in Page 3, Main text]

■ **Modified sentence**

Previously, such an exponentially decaying run distribution was theoretically elucidated within the tug-of-war model of the motor-driven cooperative transport dynamics of cargos on microtubules [38].

Comment 2-3:

At the end of Page 4, to reconcile the discrepancy between Figs. 2e and 3b, the authors introduce the aging effect, which is then demonstrated and compared with experimental data using numerical simulation. To be more comprehensive and more convincing, the authors may consider the following experimental-data-based test: From all trajectories, create a subset containing those that have at least one run phase during the observation. For each trajectory in the subset, reset the time axis so that the observation starts from the first time frame of the first run phase. After this “alignment” step, re-calculate EA MSD and compare with TA MSD. If the aging effect exists, the above operation should reverse it. Hence, we are supposed to see from the realigned experimental data the inequivalence between the EA and TA MSDs as in Fig. 3b.l.

Authors’ reply:

This is indeed a clever idea exploring the agingness in the mRNP motion in a different way. Based on the procedure suggested we analyzed the kymograph trajectories. In accordance with the Reviewer's prediction, the realigned trajectory data, having at least one run event and resetting $t=0$ to the moment the first run occurs in each trajectory, produced non-aging EA and TA MSD curves (Supplementary Fig. 13). This result is consistent with our generalized Lévy walk model with $t_a=0$ in Fig. 3b, in which the MSD curves show weak ergodicity breaking (WEB). Thus, the overlapped aged EA & TA MSD curves (in Fig. 2e) are separated and each curve displays its own distinct scaling behavior. This result is presented as one of the additional data analyses cross-checking the aging motion in Supplementary Fig. 13 with further discussion in the Figure Caption. In the main text, this is briefly mentioned as below.

[15th line, 2nd column in Page 7 to 2nd line, 1st column in Page 8, Main text]

■ **Added sentence**

Further analyses on the agingness in the mRNP transport are provided in Supplementary Figs. 13 and 14.

Comment 2-4:

In the Discussion part of the paper (at the end of Page 7), besides discussing the microscopic mechanism of the rest-time distribution, the authors may also want to discuss the microscopic mechanism of the run-time distribution.

Authors' reply:

Following the Reviewer's suggestion, in page 8 in the Discussion section we added a few sentences about the speculation of the microscopic mechanism of the exponentially truncated run distribution. Additionally, as informed in comment 2-2, we presented a new experiment suggested by Reviewer 4 about the role of the microtubule on the run phase which is substantially related to this comment. The result is provided in Supplementary Fig. 4 with an explanation in the main text (28th to 33rd line, 2nd column in Page 3).

[28th to 35th line, 1st column in Page 8, Main text]

■ **Added sentences**

Additionally, regarding the microscopic mechanism for the exponentially truncated run distribution, our control experiment demonstrated that run is a microtubule-dependent directed motion (Supplementary Fig. 4). It is speculated that the exponential truncation is related to the microtubule-associated kinesin/dynein-driven transport in a tug-of-war manner [38].

Comment 2-5:

According to the Instructions of Nature Communications (https://images.nature.com/full/nature-assets/ncomms/pdf/submission_guide_ncomms.pdf), the authors should provide a Data Availability statement in the Methods section under “Data Availability”.

Authors’ reply:

We uploaded our data in the figshare repository (doi:10.6084/m9.figshare.5549290), and added Data Availability statement in the revised manuscript.

[10th to 13th line, 2nd column in Page 9, Main text]

■ **Added sentences**

Data availability

The data sets generated and analyzed during the current study are available in the figshare repository, doi:10.6084/m9.figshare.5549290.

Reviewer #3

In this paper the Authors report experimental and theoretical evidence that the active transport dynamics of hippocampal neuronal mRNP's follows an aging Levy walk. Suppose we begin observations of a system which was initialized at $t = 0$ after some time t_a . There can be many reasons why we would want to do this, from technical restrictions of a measuring device to sheer curiosity. In many cases the delay largely changes statistical properties of the observed process. Such a phenomenon is called aging, a term which was originally used in the area of glassy materials [1] was also reported for blinking nanocrystals [2]. In such systems statistical properties (like probability distribution, mean square displacement, first-passage time) depend on a time span t_a between the initialization and the beginning of observations.

[1] E. M. Bertin and J. P. Bouchaud, *Phys. Rev. E* 67, 026128 (2003).

[2] X. Brokmann, J.-P. et al. *Phys. Rev. Lett.* 90, 120601 (2003).

Authors’ reply:

In the revised manuscript, we added the above two papers ([1] and [2]) with Ref number [49] and [50] for introducing aging phenomena in nature.

[37th to 39th line, 1st column in Page 6, Main text]

■ **Modified sentence**

A key point to reconcile this discrepancy is the aging effect shown for non-stationary non-equilibrium processes including the neuronal mRNP transport [30, 45, 49, 50].

For example, in a paper [3] aging properties of ballistic Levy’ walks and two closely related jump models: wait-first and jump-first were studied theoretically. Moreover, their probability distributions and MSDs were explicitly calculated. It turns out that despite similarities these models react very differently to the delay t_a . Aging weakly affects the shape of probability

density function and MSD of standard Levy walks.
[3] M. Magdziarz and T. Zorawik, *Phys. Rev. E* 95 022126 (2017).

Authors' reply:

We thank the Reviewer for informing us this interesting and important paper [3] on a new type of aging Lévy walk processes. This paper is introduced in the revised manuscript in page 5 with Ref number 42.

[21st to 23rd line, 2nd column in Page 5, Main text]

■ Added sentence

Recently, continuous time random walk models generalizing the conventional Lévy walks were studied [42].

Here, Authors clearly demonstrated that their model of Levy walk including rests reproduces the experimentally extracted key dynamic characteristics of mRNPs with quantitative accuracy. Moreover, this predictive theoretical model offers insight into the active target search mechanism of mRNP particles in vivo. The paper, as well as, Supplementary Information are very well written and statistically correct. According to my knowledge the paper should be interested to a wide readership since the main conclusion is the following: the proposed here new model in the form of aging Levy walk is well fitted to the extensive experimental data and distinct from the previously reported in the literature motor-driven transport. Below are a two specific comments for the consideration of the Authors.

Authors' reply:

We deeply thank the Reviewer for the very positive assessment of our work with valuable comments and for supporting it for publication. We revised the manuscript accordingly and answer the two specific comments below.

Comment 3-1:

The Authors should compare their model with those studied in paper [3]. For example, the empirical density presented on Fig.4 e corresponds to the general theoretical result presented on Fig. 4 in [3] for aging jump first Levy walk? In the manuscript Authors compare only experimental data with simulations of the fitted truncated Levy walk with rests, see Fig.3a. Following [3] one can compare the data with known theoretical density of the proper aging Levy walk.

Authors' reply:

Following the Reviewer's comment, we compared the aging jump-first model ([3]; Ref. [42] in our manuscript) with our experimental data and our generalized Lévy walk model. In paper [3], the analytic form of the aged probability density $P(x,t,t_a)$ for the jump-first model is

explicitly given as Eq. 23 in that paper. Thus, we obtained the fit-free $P(x,t;t_a)$ curve for this model with the experimentally determined values of t , t_a , and the power-law index of waiting times $\alpha=0.32$. The log-linear plot below shows the result (the same figure is provided in Supplementary Fig. 14). We found that at given parameter values the aging jump-first model explains well the contribution of rest particles (i.e., the contribution at $x=0$). (Note that in this plot the delta peak was smeared out because of the averaging within the bin size used in the experimental analysis for consistent comparison). However, we found that this model does not properly explain the experimental profile of $P(x,t)$ at x larger than the peak center. Compared to the data (and to our generalized Lévy walk), the jump-first model has a flatter profile of $P(x,t)$, meaning that the particle has a larger probability having large jumps. This is an intrinsic property of jump-first model which describes the jump and rest events with the same power-law pdf. Meanwhile, our generalized Lévy walk is controlled with two separate pdfs for the run and rest events. In our model, $P(x>30 \mu\text{m})$ should be negligible for $t>30$ s due to the finite speed of run $v\sim 1 \mu\text{m/s}$. However, the jump-first model allows a very large jump beyond this length scale due to the power-law pdf of jump events, thus having a significant probability of large displacements. For the quantitative description of mRNP transport with a Lévy walk model, we believe that the proper model needs two distinct pdfs where the rest is of a power-law while the run is a truncated pdf. We added this discussion in Supplementary Fig. 14 and its Caption.

Comment 3-2:

The Authors studied weak ergodicity breaking of their data see Fig.3.b and Fig.3C by comparing TA MSD with EA MSD. Their conclusions are right. Similar problem was studied

recently in [4], where the authors present the experimental and data analysis investigation of the weak ergodicity breaking phenomenon revealed in the dynamics of membrane proteins on the somatic surface of hippocampal neurons. A different statistical analysis and tools for the determination of ergodic properties of single particle trajectories were used there.
[4] D. Krapf et al., *Scientific Reports* 7, 5404 (2017).

Authors' reply:

The referred paper [4] indeed looks like an interesting work reporting WEB in the “subdiffusive” dynamics of a membrane protein in a hippocampal neuron. While the membrane system in [4] is different from our mRNP system, we agree that this recent paper is worth being included as a reference in the main text for explaining WEB in the sub-cellular dynamics. It is cited in page 6 in the main text with Ref number 48.

[28th to 31st line, 1st column in Page 6, Main text]

■ Modified sentence

This process is shown to exhibit WEB where the time-averaging produces $\overline{x^2(\tau)} \sim \tau/T^{1-\alpha}$ albeit the underlying motion is subdiffusive with $\langle x^2(t) \rangle \sim t^\alpha$ [30, 47, 48].

Reviewer #4

Comment 4-1:

The study by Song et al. examines the motion of b-actin mRNPs in living neurons and sets out to assign a physical model to their type of motion. The proposed model is called Levy walk. It would have been helpful if they would have tested some inhibitors of relevant motors or disassemble microtubules, to give biological relevance to the study. Citations of key studies are missing, such as: Yoon et al. Glutamate-induced RNA localization and translation in neurons. PNAS 2016

Authors' reply:

As the Reviewer suggested, we added a new control experiment data in Supplementary Fig. 4 to study the effect of nocodazole, a drug that depolymerizes microtubules, on the population of run events. We found that the percentage of mRNPs exhibiting any active (i.e., directed) motion during 1-minute observation decreased from 16% to 1% after treatment with nocodazole (5 $\mu\text{g/ml}$) for 4 hours. This experiment demonstrates that intact microtubules are indeed required for directed motion of mRNPs (i.e., the occurrence of run phases). We added a few sentences explaining this result in the main text.

[28th to 33rd line, 2nd column in Page 3, Main text]

■ Added sentences

In addition, we treated neurons with nocodazole to disrupt microtubules and found that the percentage of mRNPs with run phase was significantly decreased (Supplementary Fig. 4). This result suggests that intact microtubules are needed for run phase of mRNP movements.

We also added the above suggested paper by Yoon et al. as a new reference (Ref. [15]) in page 1 in the introduction in the main text.

[26th to 29th line, 1st column in Page 1, Main text]

■ **Modified sentence**

While mRNPs in bacteria or fibroblasts exhibit mostly diffusive or subdiffusive motion [11, 14, 16], neuronal mRNPs show distinct directed motion along the dendrites [4, 15].

REVIEWERS' COMMENTS:

Reviewer #2 (Remarks to the Author):

The authors have addressed all my concerns. I suggest accepting the manuscript for publication.

Reviewer #3 (Remarks to the Author):

I am satisfied with the Authors improvements of the revised manuscript.

Reviewer #4 (Remarks to the Author):

I am satisfied with the revision of the paper.

Add details of the nocodazole treatment to the Methods section.

Journal: Nature communications

Manuscript: NCOMMS-17-22139B

Title: Neuronal messenger ribonucleoprotein transport follows an aging Lévy walk

Authors: Minh S. Song, Hyeongseok C. Moon, Jae-Hyung Jeon, and Hye Yoon Park

Response to the Referees/Editor

We deeply thank all of our referees for their acceptance of our manuscript for publication.

Below we shortly reply to their comments.

REVIEWERS' COMMENTS:

Reviewer #2 (Remarks to the Author):

The authors have addressed all my concerns. I suggest accepting the manuscript for publication.

Reviewer #3 (Remarks to the Author):

I am satisfied with the Authors improvements of the revised manuscript.

Reviewer #4 (Remarks to the Author):

I am satisfied with the revision of the paper.

Add details of the nocodazole treatment to the Methods section.

Authors' reply:

We are pleased to find that all of the Reviewers are now satisfied with our revised manuscript and accept the manuscript. Upon the request of Reviewer #4 above, we added the details of the nocodazole treatment to the Methods section as follows:

[47th to 49th line, 2nd column in page 8, Main text]

■ **Added sentence**

For nocodazole experiments, the same dendrites were imaged first in HBS and after the treatment with 5 µg/ml nocodazole for 4 hours.